# Secure Decision Fusion in ISAC-Oriented Distributed Wireless Sensing Networks with Local Multilevel Quantization

**Guomei Zhang** [1,2,*] , **Hao Sun** [2] and **Jiayue Yu** [2]

1   State Key Laboratory of Geo-Information Engineering, Xi'an 710054, China
2   School of Information and Communications Engineering, Xi'an Jiaotong University, Xi'an 710049, China
*   Correspondence: zhanggm@mail.xjtu.edu.cn

**Abstract:** Distributed deployment for integrated sensing and communication (ISAC) can improve the sensing accuracy by exploring spatial diversity for covering the target state. However, secure fusion and limited energy consumption are still challenges for wireless-transmission-based distributed ISAC. In this paper, a secure decision-fusion scheme under energy constraint is proposed. First, the local likelihood ratios (LRs) of the local observations at sensing nodes are quantified at multiple levels corresponding to a multiple phase-shift keying (MPSK) constellation, in order to retain more sensing information. Second, an antieavesdropping scheme, which randomly rotates the constellation based on the main channel information between the nodes and ally fusion center (AFC), is proposed to confuse the data fusion of the eavesdropping fusion center (EFC). In addition, the local quantization thresholds and the rotating threshold are optimized to realize the perfect security under energy constraint and maximum rotation angle of $\pi$. In addition, the optimized rotation angle is discussed under a relaxed security requirement of the EFC in exchange for reducing the AFC error. Performance evaluation results show that the AFC error probability of the proposed scheme with a two-bit quantization and soft fusion outperforms the single-bit case and three-bit case by above 3 dB and about 0.5 dB at the error probability of $10^{-2}$, respectively. The former gain is just contributed by the more local information kept with two-bit against single-bit quantization. However, for the three-bit case, the advantage of more levels of quantization is eliminated by the worse transmission of denser constellation over a noisy channel. Moreover, the proposed scheme outperforms the conventional channel-aware encryption method under a stricter energy constraint and higher signal noise ratio (SNR).

**Keywords:** distributed wireless sensing; physical layer security; energy constraint; constellation rotation; perfect security

## 1. Introduction

In order to meet the requirement of human-centric applications, such as smart home, intelligent cities and remote healthcare, ultralow-latency- and ultrafast-data-speed-based wireless connectivity and precision sensing capability are both required in beyond-fifth-generation (B5G) and sixth-generation (6G) wireless networks [1,2]. In order to realize service-aware access and context-aware environment monitoring, sensing service will play a more significant role than ever before [1,3,4]. Integrated sensing and communications (ISAC) can combine sensing and communication functionalities in one hardware platform and is expected to utilize the congested frequency and hardware resources more efficiently [5]. Actually, wireless sensing and wireless communication have a similar evolving direction, such as higher-frequency bands, larger antenna arrays and higher energy efficiency; therefore, wireless infrastructures can be directly used to naturally implement sensing in future networks [1,6]. Then, ubiquitous sensing services to measure surrounding environments based on wireless sensing can be provided through ISAC. Wireless sensing networks (WSNs) [7–10] will be an important technology for various ISAC application

scenarios, including remote sensing, environmental monitoring, smart manufacturing and the smart Internet of things (IoT) [8,11–13].

Distributed deployment including more transmitting and receiving devices is an important type of ISAC system. It can collect information about the target from various spatial locations and obtain a better sensing performance by exploring spatial diversity [5]. In an ISAC system with distributed deployment, there are a large number of distributed sensing nodes that communicate with relays or directly with data fusion centers (FC) through wireless channels [8,12,14]. Whether to the traditional wireless sensing nodes or to the ISAC nodes, energy and frequency resources are always constrained and this is still a large challenge faced by a WSN. Therefore, distributed detection with low bandwidth and power requirement at sensing nodes is a significant issue in WSNs [15,16]. However, due to the broadcasting nature of the wireless transmission from sensing nodes to FC, there exist many security challenges [17]. Improved security is still an important network requirement of 6G [6]. Passive eavesdropping is a representative security threat faced by wireless sensing. An EFC passively overhears the signals transmitted by the sensing nodes to an AFC and also attempts to detect the target state correctly [17–19].

There are many secure solutions at various layers that can handle the eavesdropping problems. However, the physical-layer security methods with little or no aid from an encryption key and with low computational complexity is more preferable for WSNs [20–23]. A kind of key-based probabilistic ciphering schemes were proposed in [20,21], where the sensing nodes' observation was randomly mapped to a set of quantization levels according to an optimal mapping probabilities matrix. There, the mapping probabilities matrix was just the secret key. However, these works did not discuss the energy efficient issue and the effect of the transmission channel on security. Moreover, these schemes worked well under the assumption that the key was unknown by the eavesdropper. However, how to ensure this condition is satisfied was not discussed and the practicability was limited. In [22], an optimal sensing node's censoring scheme with a perfect secrecy and energy constraint was presented. However, an EFC with limited processing capability compared with an AFC was assumed and the application was still constrained. In [23], a falsified censoring strategy was proposed for the distributed detection of sparse signals. However, some extra trustworthy nodes were needed to send falsified data cooperatively in order to confuse the eavesdropper, and the hardware cost increased. Furthermore, the methods based on sending artificial noise by partial nodes were also reviewed to improve the security of distributed detection [24]. However, the extra energy would be spent to interfere with the edges and this is challenging for a WSN with limited power.

Another category of schemes consists of channel-aware encryption methods [25]. The sensing nodes decide whether to flip the bits to be transmitted according to the amplitude of the main channel between the sensing nodes and the AFC. There, the channel gains can be estimated by the sensing nodes based on the assumption that the channel is reciprocal and a pilot signal is sent by the AFC. These kinds of methods are key-free and the energy can be saved by setting some silent sensing nodes at the transmission stage. However, an efficient optimization method for some key thresholds is not given. When choosing silent sensing nodes to satisfy the energy constraint, the quality of the local decision is not taken into account. Considering the above solutions' drawbacks, we proposed a hybrid, secure, distributed detection scheme jointly considering the local decision accuracy and the wireless transmission channel in [26], which was named the joint local decision and wireless transmission (JLDWT) method. However, this work only assumed that the binary hard decision was adopted in a sensing node's local quantization. Then, a randomly bit-flipping scheme similar to that in [25] was introduced to guarantee the transmission security. The more general case of multibit quantization, which can retain more information of local observations and may get better fusion detection performance was not discussed in [26], just as in most of the related existing works [27,28]. In particular, for the sensing-node-merging communication functionality, the high-order modulation was more general to realize efficient transmission. Therefore, from these two perspectives, namely getting accurate

fusion results and matching the general communication setup, the multibit quantization of the sensing results needs to be studied. Then, to design a secure transmission scheme suitable to multibit quantization is further necessary. What is more, a perfect secrecy was assumed in [25,26]. However, in some scenarios, the better performance of the AFC is preferred and the performance constraints over the EFC can be relaxed appropriately [29]. How to realize such a flexible secure scheme is also worth studying.

The work in this paper was motivated by the above-related works and discussion. First, an antieavesdropping scheme for an energy-constrained distributed WSN is studied under the scenario that multilevel quantization is implemented over the LRs of local observations. Second, a security transmission based on a randomly rotating constellation, which is suitable for multilevel quantization, is presented under the assumption that the EFC can execute the same fusion procedure as the AFC. The abilities of the EFC are not weakened. Third, the key system parameters used in quantization and transmission, including the local quantization thresholds and the rotating threshold, are optimized to satisfy the energy constraint and the given security constraint. The rotation angle is also discussed to realize a more flexible security objective. The main contributions of this paper are summarized as follows:

(1) We propose a secure distributed decision-fusion scheme combining a multibit-quantization-based local decision and constellation-rotation-based transmission under an energy constraint for the ISAC-oriented distributed WSN. For the local decision, the multibit quantification of local LRs based on an M-PSK constellation is designed. Moreover, the detection probabilities for various decision results are derived. In particular, we let the nodes with the LRs close to one keep silent in the transmission time slot for meeting the energy constraint. For the wireless transmission, the constellation is randomly rotated according to the main channel state. The statistical independence of the main channel to the eavesdropping channel is explored to realize the security. For the data fusion, a soft fusion scheme based on the statistical information of the channel amplitude and a hard fusion scheme based on the instantaneous channel amplitude are designed, respectively.

(2) We derive the local decision thresholds of multibit quantization and the rotating threshold to realize perfect secrecy under a given power constraint and the maximized rotation angle $\pi$ for the soft fusion scheme. Next, the optimization of the rotation angle with a flexible constraint on the EFC's error performance is also discussed in relation to the soft fusion case. Through relaxing the security condition, the AFC's performance could be improved to satisfy the higher requirement from the AFC's detection performance in some scenarios.

(3) We evaluate the proposed schemes under various system conditions through a simulation. The simulation results demonstrate that the proposed schemes with two-bit quantization can obtain the best error performance of the AFC among all the schemes, especially under a relatively severe energy constraint and high SNR environment.

The rest of the paper is organized as follows: Section 2 describes the system model. Section 3 presents the proposed secure decision-fusion schemes. Section 4 discusses the optimization of the local quantification thresholds and constellation rotation angle. We give and analyze the simulation results in Section 5. Section 6 concludes the paper.

## 2. System Model

An energy constrained ISAC system with a distributed deployment shown in Figure 1 was considered, where the sensing nodes also acted as communication devices and we focused on the secure decision-fusion problem for distributed sensing in this paper. A number of sensing nodes were distributed near the physical system to detect a binary target state $\theta_0$ or $\theta_1$. Then, they quantified the local sensing results into the symbols from an M-PSK constellation. On one hand, such processing could match the general requirement of variable-order modulation for communication. On the other hand, it could obtain softer decision results about the local observations and more information can be retained. Further, similar to [25,26], these symbols were transmitted to an AFC through a wireless

parallel access channel (PAC). At the same time, an EFC passively listened to the signals between the sensing nodes and the AFC, in an effort to estimate the binary target state as well. The wireless channel between each sensing node and the AFC was called the main channel. The wireless channel from each sensing node to the EFC was denoted as the eavesdropping channel. Moreover, the same assumption as in [25] was included, where the EFC was located more than one-half wavelength apart from both the sensing devices and the AFC. Then, the main channel and the eavesdropping channel could be considered to be statistically independent of each other.

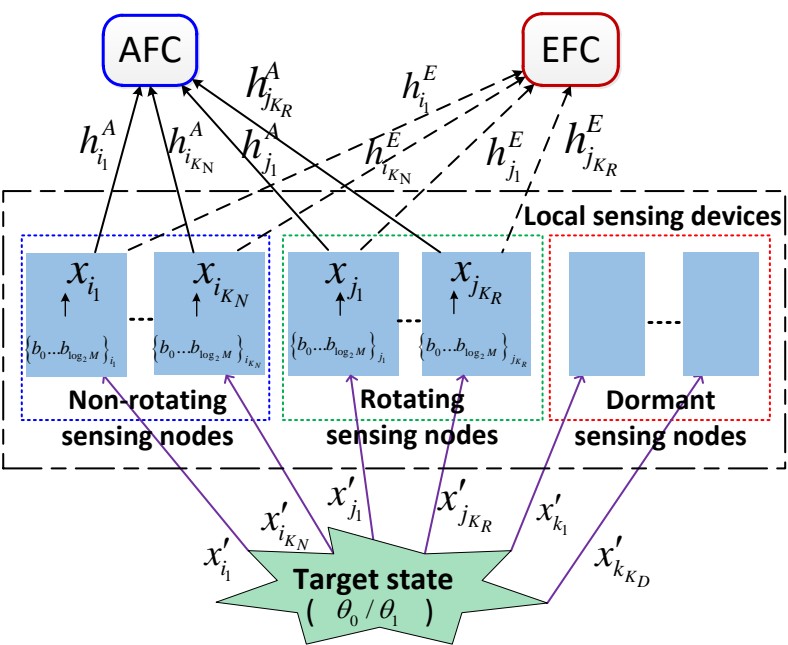

**Figure 1.** Distributed sensing system model.

In order to realize a high energy efficiency, the same idea as in the work [26] was also adopted. We let the sensing nodes be more informative to send the local decisions and otherwise dormant during the transmission phase. Moreover, among the nodes being active to send the local decisions, some sent the constellation-rotated symbols to confuse the eavesdropper in order to satisfy the security requirement. Through the above description, we can see that the sending nodes can be divided into three groups, namely the nonrotation-active group denoted as $\{i_1 \ldots i_{K_N}\}$, the rotation-active group indicated by $\{j_1 \ldots j_{K_R}\}$ and the dormant group marked with $\{k_1 \ldots k_{K_D}\}$. Obviously, the total number of sensing nodes in the network was $K = K_N + K_R + K_D$. The main channel coefficient and the eavesdropping channel coefficient were denoted by $h_k^A$ and $h_k^E$, respectively. They were independent quasi-static Rayleigh block-fading channel. We assumed $h_k^A \sim CN(0, 1)$ and $h_k^E \sim CN(0, 1)$. Moreover, the system energy constraint was equivalent to a dynamic transmission probability of total sensing nodes, denoted by $\beta$, which was proportional to the energy consumption of total sending nodes and varies from 0 to 1.

## 3. Constellation-Rotating-Based Secure Decision Fusion under Energy Constraint

### 3.1. Local Decision with Multibit Quantification

For the $k$th sensing node, the observation about the target state affected by noise was modeled as:

$$
\begin{aligned}
\theta_0: \quad & x'_k = w_k \\
\theta_1: \quad & x'_k = \theta + w_k
\end{aligned}
\tag{1}
$$

where $\theta$ is the value of a physical event, and $w_k$ is an i.i.d. zero-mean Gaussian random variable with variance $\sigma^2$, i.e., $w_k \sim N(0, \sigma^2)$. The local signal to noise ratio can be expressed as $snr_L = \theta^2/\sigma^2$. Here, it was assumed that if two prior probabilities were

identical, then the Bayesian detection could be transformed into the likelihood ratio (LR) criteria. The conditional probability distribution function of the log likelihood ratio (LLR) can be derived as [26]:

$$
\begin{aligned}
f\left(\Lambda_k^L \mid \theta_1\right) &= \frac{1}{\sqrt{2\pi \cdot snr_L}} \exp\left(-\frac{\left(\Lambda_k^L - snr_L/2\right)^2}{2 \cdot snr_L}\right) \\
f\left(\Lambda_k^L \mid \theta_0\right) &= \frac{1}{\sqrt{2\pi \cdot snr_L}} \exp\left(-\frac{\left(\Lambda_k^L + snr_L/2\right)^2}{2 \cdot snr_L}\right)
\end{aligned}
\tag{2}
$$

where $\Lambda_k^L = \log\left(\frac{f(x'_k \mid \theta_1)}{f(x'_k \mid \theta_0)}\right) = \frac{\theta}{\sigma^2} x'_k - \frac{\theta^2}{2\sigma^2}$. In order to implement the $\log_2{}^M$-bit quantization of local LRs, $M$ local detection thresholds $[\lambda_{LM/2}, ..., \lambda_{L1}, \lambda_{U1}, ..., \lambda_{UM/2}]$ were set, which satisfied $0 < \lambda_{LM/2} < ... < \lambda_{L1} \leq 1 \leq \lambda_{U1} < ... < \lambda_{UM/2} < \infty$. Each node could quantify its local observation LR to one $\log_2{}^M$ bits and then they were mapped to one symbol from the $M$-PSK constellation. This process followed the rule shown in Figure 2. In particular, the node with the likelihood ratio located between $\lambda_{L1}$ and $\lambda_{U1}$ would keep silent in the transmission time slot to save the transmitting energy.

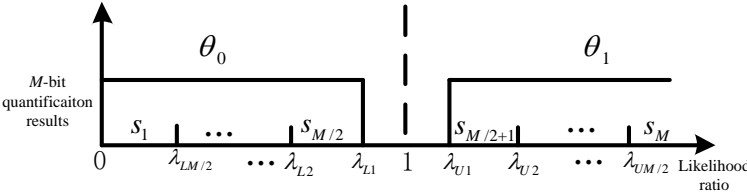

**Figure 2.** Local multiple-bit quantification.

Except for the dormant case, there were $2M$ possible cases for the local detection. They were the cases where the decision results were $M \log_2{}^M$-bit symbols in the set of $\{s_1, s_2, ..., s_M\}$ under two prior states, respectively. Then, we defined four kinds of detection probabilities, namely, the local-detection probabilities $\{P_{d_i} = P(s_{i+M/2} \mid \theta_1)\}(i \in [1, M/2])$, the missing-detection probabilities $\{P_{m_i} = P(s_{M/2-i+1} \mid \theta_1)\}(i \in [1, M/2])$, the false-detection probabilities $\{P_{f_i} = P(s_{i+M/2} \mid \theta_0)\}(i \in [1, M/2])$ and the detection probabilities under a zero prior state $\{P_{0d_i} = P(s_{M/2-i+1} \mid \theta_0)\}(i \in [1, M/2])$. Then, based on Equation (2) and Figure 2, we can derive the expressions of the above probabilities as follows:

$$
\begin{aligned}
P_{d_{M/2}} &= \int_{\log \lambda_{UM/2}}^{\infty} f\left(\Lambda_k^L \mid \theta_1\right) d\Lambda_k^L = Q\left(\frac{\log \lambda_{UM/2} - snr_L/2}{\sqrt{snr_L}}\right) \\
&\cdots \\
P_{d_i} &= \int_{\log \lambda_{Ui}}^{\log \lambda_{Ui+1}} f\left(\Lambda_k^L \mid \theta_1\right) d\Lambda_k^L = Q\left(\frac{\log \lambda_{Ui} - snr_L/2}{\sqrt{snr_L}}\right) - Q\left(\frac{\log \lambda_{Ui+1} - snr_L/2}{\sqrt{snr_L}}\right) \\
&\cdots \\
P_{m_{M/2}} &= \int_{-\infty}^{\log \lambda_{LM/2}} f\left(\Lambda_k^L \mid \theta_1\right) d\Lambda_k^L = 1 - Q\left(\frac{\log \lambda_{LM/2} - snr_L/2}{\sqrt{snr_L}}\right) \\
&\cdots \\
P_{m_i} &= \int_{\log \lambda_{Li+1}}^{\log \lambda_{Li}} f\left(\Lambda_k^L \mid \theta_1\right) d\Lambda_k^L = Q\left(\frac{\log \lambda_{Li+1} - snr_L/2}{\sqrt{snr_L}}\right) - Q\left(\frac{\log \lambda_{Li} - snr_L/2}{\sqrt{snr_L}}\right) \\
&\cdots \\
P_{f_{M/2}} &= \int_{\log \lambda_{UM/2}}^{\infty} f\left(\Lambda_k^L \mid \theta_0\right) d\Lambda_k^L = Q\left(\frac{\log \lambda_{UM/2} + snr_L/2}{\sqrt{snr_L}}\right) \\
&\cdots \\
P_{f_i} &= \int_{\log \lambda_{Ui}}^{\log \lambda_{Ui+1}} f\left(\Lambda_k^L \mid \theta_0\right) d\Lambda_k^L = Q\left(\frac{\log \lambda_{Ui} + snr_L/2}{\sqrt{snr_L}}\right) - Q\left(\frac{\log \lambda_{Ui+1} + snr_L/2}{\sqrt{snr_L}}\right) \\
&\cdots \\
P_{0dM/2} &= \int_{-\infty}^{\log \lambda_{LM/2}} f\left(\Lambda_k^L \mid \theta_0\right) d\Lambda_k^L = 1 - Q\left(\frac{\log \lambda_{LM/2} + snr_L/2}{\sqrt{snr_L}}\right) \\
&\cdots \\
P_{0di} &= \int_{\log \lambda_{Li+1}}^{\log \lambda_{Li}} f\left(\Lambda_k^L \mid \theta_0\right) d\Lambda_k^L = Q\left(\frac{\log \lambda_{Li+1} + snr_L/2}{\sqrt{snr_L}}\right) - Q\left(\frac{\log \lambda_{Li} + snr_L/2}{\sqrt{snr_L}}\right)
\end{aligned}
\tag{3}
$$

where $Q(x) = 1/\sqrt{2\pi} \int_x^{\infty} \exp\left(-t^2/2\right) dt$.

### 3.2. Random-Constellation-Rotation-Based Transmission of Local Decisions

After the local detection, the sensing nodes sent the quantified results to the AFC through a wireless PAC. Here, it was assumed that the EFC had the same prior information and processing capability as the AFC. Since the main channel and the eavesdropping channel were statistically independent, we could utilize such channel difference to realize the information transmission security in the physical layer. Referring to the same ideal of bit randomly flipping as in [25,26], the encryption based on a random constellation rotation was adopted to suit a multiple-level quantization, which randomly rotated the transmitted symbol according to the main channel state. Actually, a rotating constellation based on the main channel state was similar as the mapping diversity security scheme in [30] for the MPSK mapping. However, the latter one was proposed only for the secure wireless transmission, and it was not combined with sensing and data fusion.

Preceding the transmission, the active sensing nodes and the fusion center needed to estimate the main channel state by using the pilot symbols from the AFC and the sending nodes, respectively [25,26]. We assumed that the channel between the sensing nodes and the fusion center was reciprocal. Hence, the estimated channel information at each sensing node could be directly used to encrypt the transmitted signal, and the channel estimation results at the fusion center could be directly adopted to execute the symbols receiving and data fusing. However, since the EFC was not able to estimate the main channel state by the pilots from the AFC or the sensing nodes, it could only cancel the symbol rotation effect based on its own channel state, which was totally uncorrelated with the main channel. Then, the fusion errors had to occur randomly, and the security could be realized, even if the EFC was fully aware of the rotating thresholds and the rotating angle.

Each node determined to send the original symbol or the rotated symbol according to the main channel. There were only two transmission actions. Therefore, one threshold $t_0$ was set up and then the main channel amplitude $|h_k^A|$ was compared with $t_0$ to determine the transmission action. If the main channel gain $|h_k^A| \geq t_0$, the original constellation symbol corresponding to the local quantization value was sent. Otherwise, the rotated constellation symbol mapped to the local decision result was sent. Taking a two-bit quantization as an example, the corresponding relationship between the quantization value and the constellation symbol is given in Table 1, where $\phi$ is the rotation angle. In particular, for a rotation angle $\phi = \pi$, $s_4$ is rotated to $s_1$ and $s_1$ is rotated to $s_4$. The same condition is valid for $s_2$ and $s_3$. Extending this processing to the arbitrary $M$-PSK quantification case, we have that $s_i$ and $s_{M-i+1}$ are paired to be rotationally symmetric with each other for $i \in [1, M]$. Referring to [25,26], we selected the threshold $t_0$ to meet the conditions of $\lambda_1 = \int_{t_0}^{\infty} f(|h_k^A|) d|h_k^A| = \int_0^{t_0} f(|h_k^A|) d|h_k^A| = \lambda_2$, which meant the rotating probability was 1/2. In Section 5.1 of [26], it was proved that such a comparison threshold was one condition of perfect secrecy in the JLDWT scheme with a one-bit local quantization. In the following text, we show the same result under the multiple-bit quantization case.

**Table 1.** The relationship between the quantization value and the transmitted signal for the 2-bit case.

| Quantization | Normal Sending | Rotated Sending |
|:---:|:---:|:---:|
| $\{11\}/s_4$ | $e^{j\frac{\pi}{4}}$ | $e^{j\left(\frac{\pi}{4}+\phi\right)}$ |
| $\{01\}/s_3$ | $e^{j\frac{3\pi}{4}}$ | $e^{j\left(\frac{3\pi}{4}+\phi\right)}$ |
| $\{00\}/s_1$ | $e^{j\frac{5\pi}{4}}$ | $e^{j\left(\frac{5\pi}{4}+\phi\right)}$ |
| $\{10\}/s_2$ | $e^{j\frac{7\pi}{4}}$ | $e^{j\left(\frac{7\pi}{4}+\phi\right)}$ |

### 3.3. LLR-Based Soft Data Fusion (Sdf) at Fusion Center

Through the wireless PAC, the received signal of the AFC and EFC from the $k$th sensing node can be written as

$$z_k^A = h_k^A \cdot x_k + n_k^A$$
$$zk^E = h_k^E \cdot x_k + n_k^E \tag{4}$$

where $x_k$ is the transmitted symbol from the $M$-PSK constellation or the rotated $M$-PSK constellation. $n_k^A \sim CN(0, \sigma_A^2)$ and $n_k^E \sim CN(0, \sigma_E^2)$. Since the security scheme was based on the main channel amplitude, the influence of the channels' phase needed to be offset before data fusion. By using the pilot symbols from the sensing node, the AFC and the EFC could estimate the channel's phase. Here, we assumed the channel's phase was perfectly known by the FC. After the channel phase cancellation, the signals received could be written as

$$
\begin{aligned}
y_k^A &= e^{-j\phi_{h_k^A}} \cdot \left(h_k^A \cdot x_k + n_k^A\right) \\
y_k^E &= e^{-j\phi_{h_k^E}} \cdot \left(h_k^E \cdot x_k + n_k^E\right)
\end{aligned}
\tag{5}
$$

where $\phi_{h_k^A}$ and $\phi_{h_k^E}$ are the phases of the main channel and the eavesdropping channel, respectively. In addition, the transmission SNRs of the main channel and the eavesdropping channel are denoted as $SNR_A = |h_k^A|^2 / \sigma_A^2$ and $SNR_E = |h_k^E|^2 / \sigma_E^2$.

The received signal vector at the AFC was $\mathbf{y}^A = \left[y_1^A, y_2^A, \ldots, y_K^A\right]$ and then the fused LLR could be expressed as

$$
\Lambda^A = \frac{1}{K} \log \frac{f\left(\mathbf{y}_k^A | \theta_1\right)}{f\left(\mathbf{y}_k^A | \theta_0\right)} = \frac{1}{K} \sum_{k=1}^{K} \log \frac{f\left(y_k^A | \theta_1\right)}{f\left(y_k^A | \theta_0\right)}
\tag{6}
$$

where $f\left(y_k^A | \theta_i\right)$ is the likelihood function of the received signal from the $k$th node. According to a similar derivation as in [25,26], it can be obtained that:

$$
\begin{aligned}
f\left(y_k^A | \theta_i\right) &= \sum_{\alpha_k} \sum_{x_k} \int_0^\infty f\left(y_k^A, h_k^A, x_k, \alpha_k | \theta_i\right) dh_k^A \\
&= \sum_{\alpha_k} \sum_{x_k} \int_0^\infty f\left(y_k^A, h_k^A, x_k | \alpha_k, \theta_i\right) p(\alpha_k | \theta_i) dh_k^A \\
&= \sum_{\alpha_k} p(\alpha_k | \theta_i) \sum_{x_k} \int_0^\infty f\left(y_k^A | h_k^A, x_k, \alpha_k, \theta_i\right) f\left(h_k^A, x_k | \alpha_k, \theta_i\right) dh_k^A \\
&\overset{(a)}{=} \sum_{\alpha_k} p(\alpha_k | \theta_i) \sum_{x_k} \int_0^\infty f\left(y_k^A | h_k^A, x_k\right) f\left(h_k^A\right) p(x_k | \alpha_k) dh_k^A \\
&\overset{(b)}{=} \sum_{m=1}^{M} p(\alpha_k = s_m | \theta_i) \left[\int_{t_0}^{+\infty} f\left(y_k^A | h_k^A, x_k = s_m\right) f\left(h_k^A\right) dh_k^A\right. \\
&\qquad \left. + \int_0^{t_0} f\left(y_k^A | h_k^A, x_k = s_m e^{j\phi}\right) f\left(h_k^A\right) dh_k^A\right] \\
&\qquad + p\left(\alpha_k = null | \theta_i\right) \int_0^{+\infty} f\left(y_k^A | h_k^A, x_k = 0\right) f\left(h_k^A\right) dh_k^A \\
&= \sum_{m=1}^{M} p(\alpha_k = s_m | \theta_i) \left[\Phi\left(t_0, \infty, s_m, y_k^A, \sigma_A^2\right)\right. \\
&\qquad \left. + \Phi\left(0, t_0, s_m e^{-j\phi}, y_k^A, \sigma_A^2\right)\right] \\
&\qquad + p\left(\alpha_k = null | \theta_i\right) \Phi\left(0, \infty, 0, y_k^A, \sigma_A^2\right)
\end{aligned}
\tag{7}
$$

where

$$
\begin{aligned}
\Phi\left(t_a, t_b, x_k, y_k^A, \sigma_A^2\right) &= \int_{t_a}^{t_b} f\left(y_k^A | h_k^A, x_k\right) f\left(|h_k^A|\right) d|h_k^A| \\
&= \int_{t_a}^{t_b} \frac{1}{\pi \sigma_A^2} \exp\left(-\frac{|y_k^A - |h_k^A| x_k|^2}{\sigma_A^2}\right) \cdot 2|h_k^A| \exp\left(-|h_k^A|^2\right) d|h_k^A|
\end{aligned}
\tag{8}
$$

In Equation (7), (a) is valid because $\theta_i \to \alpha_k \to x_k \to y_k^A$ forms a Markov chain and $h_k^A$ is uncorrelated with $x_k$ and $\theta_i$. Moreover, (b) is because $p(x_k = s_m | \alpha_k = s_m) = 1$ for $|h_k^A| \geq t_0$ and $p(x_k = s_m e^{-j\phi} | \alpha_k = s_m) = 1$ for $|h_k^A| < t_0$. Obviously, the above fusion strategy is a soft one based on the statistic distribution of the channel amplitude. Here, the Rayleigh distribution $f(|h|) = 2|h| \exp\left(-|h|^2\right)$ was assumed both for $|h_k^A|$ and $|h_k^E|$. At last, the AFC used the Bayesian detection rule $\Lambda^A \overset{\theta_1}{\underset{\theta_0}{\gtrless}} 0$ to get the final data fusion result.

At the EFC, the same fusion processing as in Equations (6) and (7) was adopted except that $(y_k^A, h_k^A, \sigma_A^2)$ were replaced by $(y_k^E, h_k^E, \sigma_E^2)$.

In order to analyze the error probability at the AFC, the statistic distribution of $\Lambda^A$ is first discussed. Obviously, the received signals from different sensing nodes are independent of each other. Then, $\Lambda_k^A = \log \frac{f(y_k^A|\theta_1)}{f(y_k^A|\theta_0)}$ can be taken as the i.i.d. random variables. Further, based on the central limit theorem, we can approximately take the statistic distribution of $\Lambda^A$, which is the average of $K$ i.i.d. random variables, as a normal distribution for a large $K$, that is, $\Lambda^A|\theta_i \sim \mathcal{N}(\mu_A|\theta_i, \frac{\gamma_A^2|\theta_i}{K})$, where $\mu_A|\theta_i$ and $\gamma_A^2|\theta_i$ are the mean and variance of $\Lambda_k^A$ conditioned on $\theta_i$, respectively. Thus, according to the Bayesian detection rule, the final error probability of AFC can be expressed as

$$
\begin{aligned}
P_e^A &= q_0 P(\Lambda^A \geq 0|\theta_0) + q_1 P(\Lambda^A < 0|\theta_1) \\
&= q_0 \cdot [1 - Q(\frac{\mu_A|\theta_0}{\sqrt{\gamma_A^2|\theta_0/K}})] + q_1 \cdot Q(\frac{\mu_A|\theta_1}{\sqrt{\gamma_A^2|\theta_1/K}})
\end{aligned}
\tag{9}
$$

where $\mu_A|\theta_i = \int_{-\infty}^{\infty} \Lambda^A f(\Lambda^A|\theta_i) d\Lambda^A$ and $\gamma_A^2|\theta_i = \int_{-\infty}^{\infty} (\Lambda^A)^2 f(\Lambda^A|\theta_i) d\Lambda^A - (\mu_A|\theta_i)^2$.

### 3.4. Hard-Decision-Based Hard Data Fusion (Hdf) at Fusion Center

If the instantaneous channel amplitude from the sensing node to the FC could be estimated and used, the hard decision could be executed at the AFC and the EFC. We rewrote Equation (5) as:

$$
\begin{aligned}
y_k^A &= |h_k^A| x_k + e^{-j\phi_{h_k^A}} n_k^A \\
y_k^E &= |h_k^E| x_k + e^{-j\phi_{h_k^E}} n_k^E
\end{aligned}
\tag{10}
$$

For the sensing nodes keeping dormant at the transmission stage, the corresponding received signals of these nodes were invalid and needed to be excluded at the fusion stage. Since the received signal only included the noise for the dormant nodes, a small threshold $\varepsilon$ was set up and combined with the noise variance to realize the valid signal selection. If $|y_k^A|^2 > \sigma_A^2 + \varepsilon$, $y_k^A$ was retained in the hard decision and data fusion. We assumed the channel amplitude information was perfectly known by the AFC, and the decision processing was expressed as

$$
s_k^* = \begin{cases} \underset{s_k \in \{s_1,...,s_M\}}{\arg\min} \ |y_k^A - |h_k^A| s_k|^2 & |h_k^A| \geq t_0 \\ \underset{s_k \in \{s_1,...,s_M\}}{\arg\min} \ |y_k^A - |h_k^A| e^{j\phi} s_k|^2 & |h_k^A| < t_0 \end{cases}
\tag{11}
$$

Further, if $s_k^* \in \{s_1, ..., s_{M/2}\}$, the decision result was $\theta_0$. Otherwise, it was $\theta_1$. Denoting the number of the valid received signals as $K_v$ and the number of decision results being $\theta_1$ as $N_{\theta_1}$, the hard data fusion result was given by

$$
\theta_F = \begin{cases} \theta_1, & N_{\theta_1} > K_v/2 \\ \theta_0, & otherwise \end{cases}
\tag{12}
$$

## 4. Optimization of Local Quantification and Constellation Rotation

### 4.1. Perfect Security Analysis

In this subsection, the condition of perfect secrecy for the proposed secure decision fusion scheme with multiple-level quantization is analyzed. Perfect secrecy means that the EFC's fusion results have the same probability for both cases of $\theta_0$ and $\theta_1$. That also implies the error probability at the EFC almost stays at 0.5. Considering the maximum rotation angle $\pi$ of the $M$-PSK constellation, which moves each constellation point to its rotationally symmetric point and may deduce the most serious error of the EFC, we took such a rotation angle as one condition of perfect secrecy in our scheme. Let us begin with the conditional likelihood function of the $k$th sensing node obtained by the EFC to derive

the optimal local quantization thresholds under the perfect secrecy constraint. Referring to Equation (7), it can be obtained that

$$
\begin{aligned}
f(y_k^E|\theta_i) &= \sum_{\alpha_k} p(\alpha_k|\theta_i) \sum_{x_k} \int_0^\infty f(y_k^E|h_k^E, x_k) f(h_k^E) p(x_k|\alpha_k) dh_k^E \\
&\overset{(a)}{=} \sum_{m=1}^M p(\alpha_k = s_m|\theta_i) \Big[ \int_0^{+\infty} f(y_k^E|h_k^E, x_k = s_m) f(h_k^E) dh_k^E \cdot \int_{t_0}^{+\infty} f(h_k^A) dh_k^A \\
&\qquad + \int_0^{+\infty} f(y_k^E|h_k^E, x_k = s_m e^{j\phi}) f(h_k^E) dh_k^E \cdot \int_0^{t_0} f(h_k^A) dh_k^A \Big] \\
&\qquad + p(\alpha_k = null|\theta_i) \int_0^{+\infty} f(y_k^E|h_k^E, x_k = 0) f(h_k^E) dh_k^E \\
&\overset{(b)}{=} \sum_{m=1}^M p(\alpha_k = s_m|\theta_i) \Big[ \Phi(0, +\infty, s_m, y_k^E, \sigma_E^2) \cdot \eta \\
&\qquad + \Phi(0, +\infty, s_m e^{j\phi}, y_k^E, \sigma_E^2) \cdot (1 - \eta) \Big] \\
&\qquad + p(\alpha_k = null|\theta_i) \Phi(0, +\infty, 0, y_k^E, \sigma_E^2)
\end{aligned}
\tag{13}
$$

where (a) is still due to the conditions that $p(x_k = s_m|\alpha_k = s_m) = 1$ for $|h_k^A| \geq t_0$ and $p(x_k = s_m e^{-j\phi}|\alpha_k = s_m) = 1$ for $|h_k^A| < t_0$, and $h_k^A$ is independent from $(y_k^E|h_k^E, x_k)$. The definition $\eta \overset{\Delta}{=} \int_{t_0}^{+\infty} f(h_k^A) dh_k^A$ makes (b) hold. In the case where the rotation angle $\phi = \pi$, we have $s_m = s_{M-m+1} e^{j\pi}$. Therefore, Equation (13) can be rewritten as

$$
\begin{aligned}
f(y_k^E|\theta_i) &= \sum_{m=1}^M [p(\alpha_k = s_m|\theta_i)\eta + p(\alpha_k = s_{M-m+1}|\theta_i)(1 - \eta)] \cdot \Phi(0, +\infty, s_m, y_k^E, \sigma_E^2) \\
&\quad + [1 - \sum_{m=1}^M [p(\alpha_k = s_m|\theta_i)] \Phi(0, +\infty, 0, y_k^E, \sigma_E^2)
\end{aligned}
\tag{14}
$$

To realize perfect security, it is required that $f(y_k^E|\theta_1) = f(y_k^E|\theta_0)$. Combined this condition with (14), we obtain

$$
\begin{aligned}
p(\alpha_k = s_m|\theta_1)\eta &+ p(\alpha_k = s_{M-m+1}|\theta_1)(1 - \eta) \\
&= p(\alpha_k = s_m|\theta_0)\eta + p(\alpha_k = s_{M-m+1}|\theta_0)(1 - \eta) \quad \forall m \in [1, M] \\
\sum_{m=1}^M p(\alpha_k = s_m|\theta_1) &= \sum_{m=1}^M p(\alpha_k = s_m|\theta_0)
\end{aligned}
\tag{15}
$$

Further, solving Equation (15), we get

$$
\begin{aligned}
\eta &= 1/2 \\
p(s_m|\theta_1) + p(s_{M-m+1}|\theta_1) &= p(s_m|\theta_0) + p(s_{M-m+1}|\theta_0) \quad \forall m \in [1, M]
\end{aligned}
\tag{16}
$$

The first condition in Equation (16) just means $\int_{t_0}^{+\infty} f(h_k^A) dh_k^A = \int_0^{t_0} f(h_k^A) dh_k^A = 1/2$ and further results in $t_0 = \sqrt{\log(2)}$. The rest of the conditions are combined with Equation (3) to obtain

$$
P_{d_i} + P_{m_i} = P_{f_i} + P_{0d_i} \quad \forall i \in [1, M/2]
\tag{17}
$$

Then, we need to find the local detection thresholds $[\lambda_{LM/2}, ..., \lambda_{L1}, \lambda_{U1}, ..., \lambda_{UM/2}]$ to meet the conditions in (17). Similar to Equation (19) in [26], we define the following functions

$$
\begin{aligned}
D_U(\lambda) &\overset{\Delta}{=} \int_{\log(\lambda)}^\infty [f(\Lambda_k^L|\theta_1) - f(\Lambda_k^L|\theta_0)] d\Lambda_k^L \\
D_L(\lambda) &\overset{\Delta}{=} \int_{-\infty}^{\log(\lambda)} [f(\Lambda_k^L|\theta_0) - f(\Lambda_k^L|\theta_1)] d\Lambda_k^L
\end{aligned}
\tag{18}
$$

Further, we can derive that

$$
\begin{aligned}
D_L(\lambda) &= \int_{-\infty}^{\log(\lambda)} [f(\Lambda_k^L|\theta_0) - f(\Lambda_k^L|\theta_1)] d\Lambda_k^L \\
&= -\int_{-\infty}^{\log(\lambda)} [f(\Lambda_k^L|\theta_1) - f(\Lambda_k^L|\theta_0)] d\Lambda_k^L \\
&\overset{(a)}{=} -\left(0 - \int_{\log(\lambda)}^{\infty} [f(\Lambda_k^L|\theta_1) - f(\Lambda_k^L|\theta_0)] d\Lambda_k^L \right) \\
&= D_U(\lambda)
\end{aligned}
\tag{19}
$$

where (a) is due to the fact that $\int_{-\infty}^{+\infty} [f(\Lambda_k^L|\theta_1) - f(\Lambda_k^L|\theta_0)] d\Lambda_k = 0$ based on the total probability theory.

Then, using Equations (3) and (17)–(19), we have

$$
\begin{aligned}
D_U(\lambda_{UM/2}) &= P_{d_{M/2}} - P_{f_{M/2}} = P_{0d_{M/2}} - P_{m_{M/2}} \\
&= D_L(\lambda_{LM/2}) = D_U(\lambda_{LM/2}) \\
D_U(\lambda_{Ui}) - D_U(\lambda_{Ui+1}) &= P_{d_i} - P_{f_i} = P_{0d_i} - P_{m_i} \\
&= D_L(\lambda_{Li}) - D_L(\lambda_{Li+1}) = D_U(\lambda_{Li}) - D_U(\lambda_{Li+1}) \ \forall i \in [1, \tfrac{M}{2} - 1]
\end{aligned}
\tag{20}
$$

Substituting the first formula into the second formula with $i = \frac{M}{2} - 1$ in Equation (20), we obtain $D_U(\lambda_{U\frac{M}{2}-1}) = D_U(\lambda_{L\frac{M}{2}-1})$. By the same manner, it can be deduced that Equation (20) is equivalent to

$$
D_U(\lambda_{Ui}) = D_U(\lambda_{Li}) \quad \forall i \in [1, M/2]
\tag{21}
$$

Finally, the perfect security conditions in the second row of Equation (16) are transferred to Equation (21). For the convenience of analysis, we redraw the curve of $D_U(\lambda)$ shown by Figure 3 in [26] here.

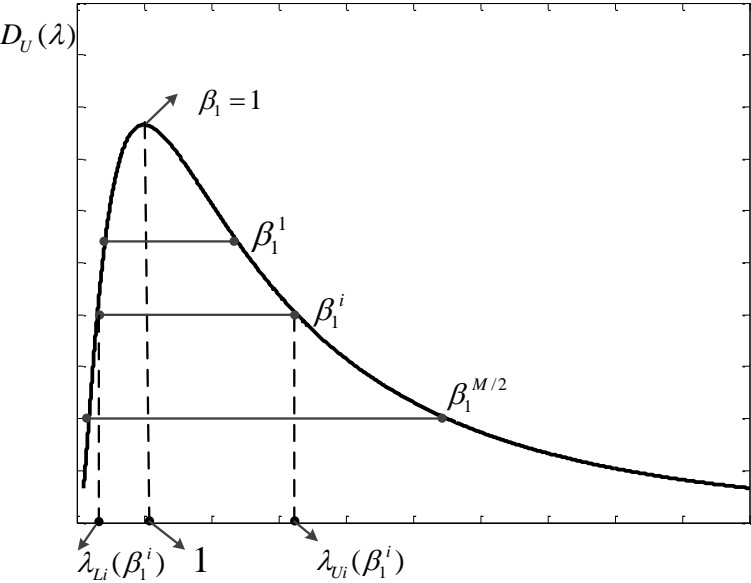

**Figure 3.** Diagram of function $D_U(\lambda)$.

In Figure 3, the X axis represents the independent variable $\lambda$ of function $D_U(\lambda)$ given in Equation (18) and the Y axis corresponds to the value of function $D_U(\lambda)$. Actually, $\lambda$ represents the likelihood ratio of the local detection. From Figure 3, it can be seen that there

is a pair of thresholds $\lambda_{Ui}$ and $\lambda_{Li}$ corresponding to one value of $D_U(\lambda)$. Moreover, this $D_U(\lambda)$ maps to a certain $\beta_1^i$ or a certain $\beta_0^i$, which are defined as

$$
\begin{aligned}
\beta_1^i &= \int_{\log(\lambda_{Ui})}^{\infty} f\left(\Lambda_k^L | \theta_1\right) d\Lambda_k^L + \int_{-\infty}^{\log(\lambda_{Li})} f\left(\Lambda_k^L | \theta_1\right) d\Lambda_k^L \\
&= \sum_{j=i}^{M/2} \left(P_{d_j} + P_{m_j}\right) \\
\beta_0^i &= \int_{\log(\lambda_{Ui})}^{\infty} f\left(\Lambda_k^L | \theta_0\right) d\Lambda_k^L + \int_{-\infty}^{\log(\lambda_{Li})} f\left(\Lambda_k^L | \theta_0\right) d\Lambda_k^L \\
&= \sum_{j=i}^{M/2} \left(P_{f_j} + P_{0d_j}\right)
\end{aligned}
\tag{22}
$$

From Equation (21), it can be easily obtained that $\beta_1^i = \beta_0^i$. Therefore, we only focus on the discussion of $\beta_1^i$. Obviously, Figure 3 shows that when $\lambda_{Ui}$ and $\lambda_{Li}$ overlap at 1, the maximum $D_U(\lambda)$ and the maximum $\beta_1^i = 1$ are obtained simultaneously. With $\lambda_{Ui}$ and $\lambda_{Li}$ keeping away from 1, $D_U(\lambda)$ and $\beta_1^i$ both decrease monotonically. If $\beta_1^i, \forall i \in [1, M/s]$ are given, the corresponded threshold pairs $\{\lambda_{Ui}, \lambda_{Li}\}, \forall i \in [1, M/s]$ can be directly determined based on the function $D_U(\lambda)$ and the condition of Equation (21) through a numerical calculation combined with searching.

Next, we discuss the determination of $\beta_1^i$. First, $\beta_1^1$ determined by $\lambda_{U1}$ and $\lambda_{L1}$ actually represents the practical transmission probability of sensing nodes, which should satisfy the constraint $\beta_1^1 \leq \beta$. According to the analysis of Section 5 in [26], we know that the minimized error probability at the AFC can be obtained when $\beta_1^1$ equals $\beta$ under a low or high-SNR range. This condition is also involved in the case of multiple-bit quantification. As for the other $\beta_1^i$, no limitation is found in terms of concrete value from the perspective of perfect security. Then, we use a very simple criterion to select $\beta_1^i, \forall i \in [2, M/2]$, that is $\beta_1^i = \beta_1^1 \cdot [1 - (i-1)/(M/2)], \quad \forall i \in [2, M/2]$. This criterion also means the probability of the quantification result falling in each interval is the same, i.e., $P_{d_i} + P_{m_i} = P_{d_j} + P_{m_j}, \forall i, j \in [1, M/2]$ and $i \neq j$.

### 4.2. Optimization of Constellation Rotation under a Given Error Constraint of EFC

In Section 4.1, we derived the local quantification thresholds and the rotating threshold under the perfect security constraint with the rotation angle being $\pi$. However, the error performance at the AFC was also reduced as perfect security is realized, to keep the worst performance at the EFC, while for the case where no such serious requirement on the security performance at the EFC was needed, it could be relaxed to some degree in exchange for a better performance at the AFC.

From the proposed scheme, we found that the rotation angle $\phi$ was an important factor for controlling the security performance. Moreover, when only the statistic channel information or the imperfect instantaneous channel state was useful at the AFC, the rotated constellation could not be recovered perfectly and then, it would inevitably have a negative response to the AFC's error performance. For a smaller $\phi$, the quantification constellation points had less distortion after rotating. The imperfect rotation offset would also cause less influence on the data fusion at the AFC. Surely, a smaller $\phi$ also meant less degree of confusion at the EFC. Hence, an optimization problem was designed to find an appropriate $\phi$ to satisfy the specified constraint of the EFC's error probability.

In order to simplify the sensing nodes' local detection and integrate various levels of security limitations, including perfect security, into a unified system, the optimization methods of local decision thresholds in Section 4.1 were kept under a certain energy constraint $\beta$. Therefore, the optimization of local quantification was decoupled from the optimization of constellation rotation. The latter one could be established as

$$
\begin{aligned}
&\min_{\phi} \quad \left| P_{th}^E - P_e^E(\phi) \right| \\
&\text{subject to}: \quad 0 \leq \phi < \pi
\end{aligned}
\tag{23}
$$

where $P_{th}^E$ is the given error constraint of the EFC. $P_e^E(\phi)$ is the actual error performance of the EFC, which is a function of $\phi$ and can also be computed by Equation (9) for the soft data fusion. From Equations (7) and (8), we can see that the relationship between $\phi$ and $f\left(y_k^E|\theta_i\right)$ was a complex weighted integral form. It was more difficult to get the analysis formulas of $\mu_E|\theta_i$ and $\gamma_E^2|\theta_i$ with respect to $\phi$. Therefore, a Monte Carlo simulation was used to obtain the sample statistics of $\mu_E|\theta_i$ and $\gamma_E^2|\theta_i$ in our simulation experiment.

Combing the two parts of the optimization from Sections 4.1 and 4.2, we conclude the optimization algorithm (Algorithm 1) of key parameters in our scheme, which is given as follows:

---

**Algorithm 1:** Optimization Algorithm of Key Parameters for Constellation-Rotating-Based Secure Decision Fusion under a Given Error Constraint of EFC

---

1  Initialization :
2  $M, \beta, \Delta\phi, \Delta\lambda, P_{th}^E, t_0 = \sqrt{\log(2)}$
3  $\phi^* = \phi_0 = 0;\ \lambda_{L0}^* = 1;\ \varepsilon_1 = 1;\ \varepsilon_2 = 0.5; K = \left\lfloor \frac{\lambda_{L0}}{\Delta\lambda} \right\rfloor; N = \left\lfloor \frac{\pi}{\Delta\phi} \right\rfloor$
4  Optimization of $\{\lambda_{Ui}, \lambda_{Li}\}, \forall i \in [1, M/2]$ :
5  $\beta_1^1 = \beta;\ \beta_1^i = \beta_1^1 \cdot [1 - (i-1)/(M/2)],\quad \forall i \in [2, M/2]$
6  For $i = 1$ to $M/2$
7      For $k = 1$ to $K$
8          $(\lambda_{Li})_k = \lambda_{L(i-1)}^* - k\Delta\lambda$
9          Find $(\lambda_{Ui})_k$ to meet $D_U((\lambda_{Ui})_k) = D_U((\lambda_{Li})_k)$
10        Calculate $(\beta_1^i)_k$ based on Equation (22)
11        If $\left|(\beta_1^i)_k - \beta_1^i\right| < \varepsilon_1$
12           $\varepsilon_1 = \left|(\beta_1^i)_k - \beta_1^i\right|$
13           $\lambda_{Li}^* = (\lambda_{Li})_k;\ \lambda_{Ui}^* = (\lambda_{Ui})_k$
14        End If
15      End For
16  End For
17  Optimization of $\phi$ :
18  For $n = 1$ to $N$
19      $\phi_n = \phi_0 + n\Delta\phi;$
20      Calculate $(P_e^E)_n$ based on Equation (9);
21        If $\left|(P_e^E)_n - P_{th}^E\right| < \varepsilon_2$
22          $\varepsilon_2 = \left|(P_e^E)_n - P_{th}^E\right|$
23          $\phi^* = \phi_n$
24      End If
25  End For
26  $\phi^*, \{\lambda_{Li}^*, \lambda_{Ui}^*\}$ are obtained.

---

## 5. Simulation Results and Discussions

In this section, the performance of the proposed random-constellation-rotation-based secure decision fusion scheme towards the local multiple-bit quantization were evaluated through a computer simulation in a distributed ISAC system with one AFC and one EFC. The simulation tool was MATLAB R2016a. Some primary simulation parameters are listed in Table 2. It is noted that all the assumptions about the EFC were the same as those for the AFC, such as the SNR of the transmission channel and the prior information of the local decision thresholds and the rotation angle. We selected two baseline schemes to be compared with the proposed one, which were the modified scheme applicable for a multiple-bit local decision based on the channel aware encryption presented in [25] and the JLDWT scheme with a one-bit local decision given in [26]. In both baseline schemes, the condition of perfect security to the EFC was assumed. For simplification, we denoted the two baseline schemes as scheme 1 and scheme 2 in the simulation result figures, respectively.

**Table 2.** Simulation parameter settings.

| Simulation Parameters | Parameter Settings |
| --- | --- |
| Channel model | Rayleigh block-fading channel |
| Number of sensing nodes | 20, 50 |
| Number of quantization bits | 2, 3 |
| Local detection SNR | $snr_L = 5$ dB |
| Transmission channel SNR | $SNR_A = SNR_E = -2 : 2 : 16$ dB |
| Energy constraint | $\beta = 0.4 : 0.1 : 1$ |

*5.1. Error Performance of Soft Data Fusion*

Figure 4 shows the error probabilities of the AFC and the EFC versus the transmission SNR for three schemes when the system energy constraint was 0.8 and the perfect security condition was satisfied. Here, the soft data fusion method based on the statistical channel amplitude information was adopted. It can be seen that the error probabilities of the EFC for all three schemes were 0.5 and the perfect security condition was met.

Focusing on the performance of the AFC, we can see that the proposed scheme with two-bit quantization obviously outperformed the two baseline schemes with one-bit quantization. At the error probability of $10^{-2}$, the SNR gain against Scheme 1 with one-bit quantization was about 4 dB and the SNR gain against scheme 2 was slightly higher than 3 dB. Actually, the error performance of the AFC mainly depended on two factors, including the local decision accuracy and the wireless transmission reliability. The local observations of all the schemes in Figure 4 were obtained by the same likelihood ratio (LR) criteria over Equation (1), but more information about the same local observation LR was kept by the two-bit quantization than in the one-bit case. It could contribute to the accuracy improvement of the final data fusion under a fixed transmission reliability. As for the effect of the transmission, from the perspective of only one node, the target state was estimated incorrectly only when both bits were wrongly received. Although, as we all know, the bit error rate of 4PSK (corresponding to two-bit quantization) is higher than 2PSK (corresponding to 1-bit quantization) under an identical symbol SNR, so the probability of two bits being wrong simultaneously was still lower than the case for one bit. This advantage was just obtained by the diversity gain of transmission over I and Q dual orthogonal channels. That is to say, for two-bit local quantization, the advantage of more local information overcame the reduction of the transmission performance.

Moreover, in the high-SNR range (above 12 dB), the proposed method with two-bit quantization also had a slight performance gain over the Scheme 1 expanded to two-bit quantization. This was due to the contribution of selecting the dormant sensing nodes based on the local decision quality. It prevented the low-quality local decisions worsening the final fusion performance. However, in the expanded version of scheme 1, only a black space was kept to prevent the confusion of rotating and nonrotating cases and its contribution to the final fusion was relatively lower. Such a gap became more obvious when the effect of the transmission errors was reduced in a higher-SNR range.

In addition, when the number of quantization bits increased from two to three, the error performance of the proposed scheme could not be improved further, while it degraded slightly, especially in the low-SNR range (below 10 dB). The same result also happened to scheme 1. This was because the effect of the transmission errors on the final fusion became dominant. For the three-bit case, the number of independent transmission channels was still two and there was no extra diversity gain. However, the antinoise ability of a higher-order modulation declined because its constellation was denser. Then, the significant degradation of transmission performance eliminated the advantage of more local levels of quantization in the final fusion, especially in the low-SNR range; this elimination phenomenon was more obvious due to a worse transmission at a low SNR. This result indicated that the two-bit quantization of the local decision was more preferable in the considered distributed detection system.

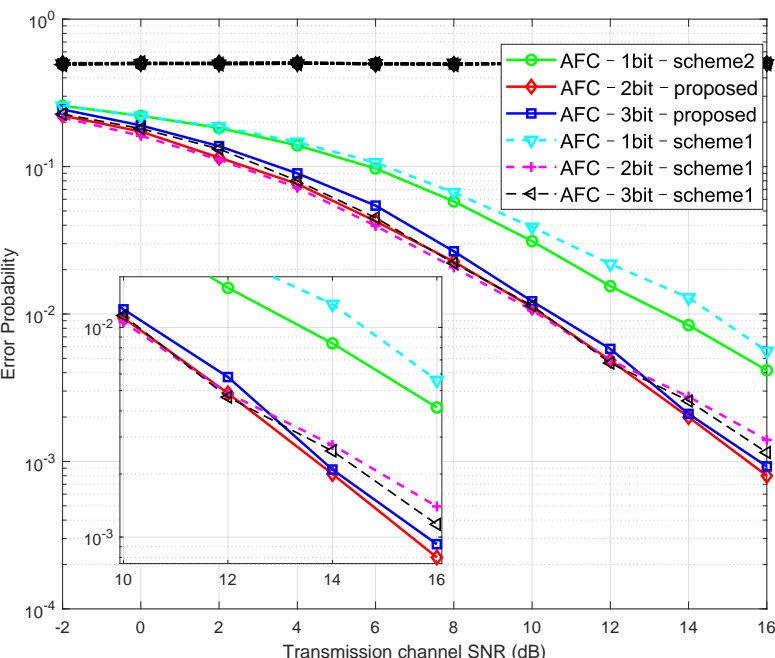

**Figure 4.** Error probabilities of the soft data fusion varying with the SNR of transmission channel for $\beta = 0.8$ and $snr_L = 5$ dB.

The error probabilities of the AFC and the EFC versus the system energy constraint $\beta$ are given in Figure 5, where transmission SNRs of 5 dB and 15 dB were considered and the perfect security condition was requested. The energy constraint $\beta$ indicated the probability of the active sensing nodes among the total nodes. It can be seen from Figure 5 that the error probabilities of the EFC for all three schemes were at 0.5, for all $\beta$'s. As for the performance of the AFC, we can see that the two-bit scheme significantly outperformed scheme 2 with one-bit quantization at all $\beta$'s. This was mainly contributed to by the additional information about the local observations at the final data fusion. Compared with Scheme 1 with a two-bit local decision, our scheme still had a significant gain at a low $\beta$ and high SNR. To be specific, for $\beta = 0.4$ and an SNR of 15 dB, the error probability of AFC decreased from about $3 \times 10^{-2}$ to $7 \times 10^{-3}$. This showed that canceling the low-quality local detection results from the fusion data was advantageous and it became more obvious for a good transmission channel. Furthermore, we can see that the AFC performance improved with $\beta$ increasing when $\beta$ was less than 0.8. However, when $\beta > 0.8$, the performance of the AFC showed a floor effect and even reduced a little with an increasing $\beta$ for the three schemes. As for scheme 2 and the proposed two-bit scheme, that was because more poor local decisions participated in the final data fusion and the fusion performance was worsened when nearly all sensing nodes (when $\beta$ was close to one) sent their local decision results to the FC, while for scheme 1, this phenomenon was induced by the confusion of the AFC about distinguishing between the rotating and nonrotating states.

Figure 6 shows the effect of constellation rotation angle $\phi$ on the error performance under transmission SNRs of 5 dB and 10 dB, where the recommended two-bit quantification scenario with soft data fusion was considered. We can see that the error probabilities of the AFC and EFC both increased with $\phi$ varying from 0 to $\pi$ for a relative low transmission SNR. In particular, the error performance of the EFC and AFC was the same and the best at $\phi = 0$. The worst performance was obtained at $\phi = \pi$ for both FCs. For $\phi = \pi$, perfect security, i.e., the error probability at the EFC holding at 0.5, was realized. This result agreed with the analysis in Sections 4.1 and 4.2. Moreover, for the relative high transmission SNR of 10 dB, the above phenomena still existed on the whole. However, the error performance no longer varied with $\phi$ monotonously in the whole range of $[0, \pi]$, while for $\phi$ falling in the

range $[0, \pi/2]$ and $[3\pi/4, \pi]$, the case of a monotonous increase was still kept. The results shown in Figure 6 just give us the idea that we can satisfy a flexible EFC error constraint and get a better error performance of the AFC by optimizing $\phi$.

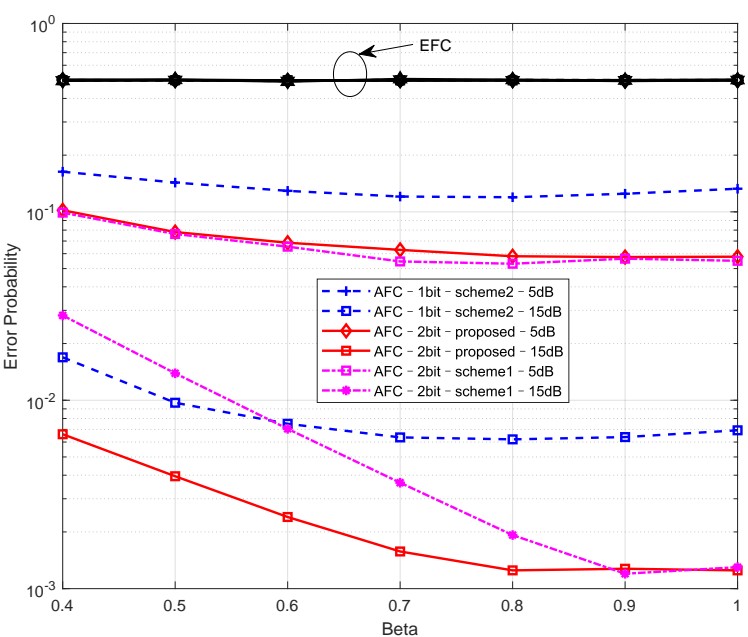

**Figure 5.** Error probabilities of the soft data fusion varying with $\beta$ for SNR = 5, 15 dB and $snr_L$ = 5 dB.

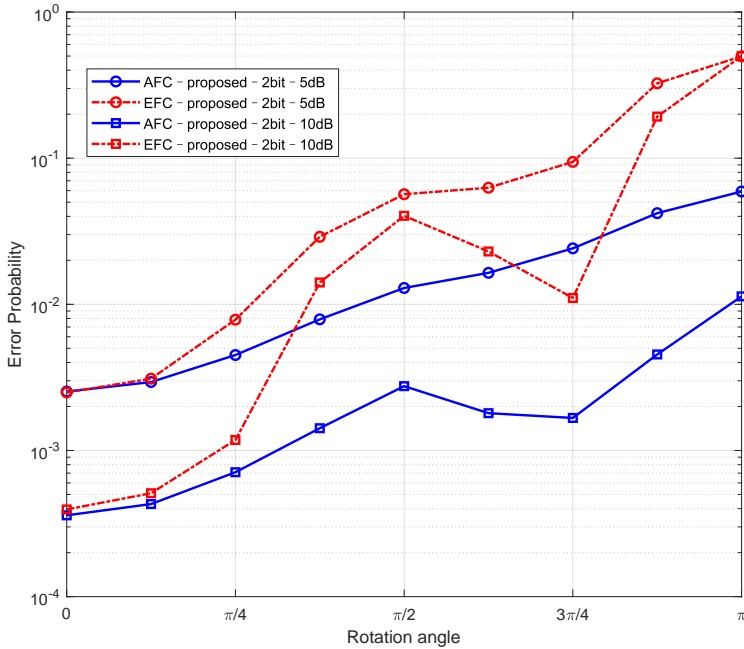

**Figure 6.** Error probabilities of the soft data fusion varying with $\phi$ for SNR = 5, 10 dB and $snr_L$ = 5 dB.

### 5.2. Error Performance of Hard Data Fusion

Figure 7–9 show the error probabilities of the hard data fusion method as functions of the transmission SNR, $\beta$ and $\phi$, respectively. Here, the perfect channel amplitude information from the sensing nodes to the fusion center was assumed to be known at the AFC and the EFC. From Figure 7, it can be seen that perfect security was still guaranteed by

the proposed scheme when a hard data fusion was utilized. Moreover, we can see that our secure scheme with two-bit local quantization was first a little worse and then a little better than the case of a one-bit local decision, with the SNR improving from the perspective of the AFC performance. This was because the degradation of the antinoise ability of the higher-order modulation became smaller for the higher SNR and the advantage of multiple-bit quantization became dominant. However, the final fusion performance decreased more obviously for three-bit quantification compared with the soft data fusion. That is to say the hard fusion was more sensitive to the transmission quality from the sensing nodes to the FCs. The higher bit error rate of 8PSK with a hard decision worsened the final fusion performance significantly compared with 2PSK and 4PSK. Therefore, either one-bit or two-bit local quantification should be selected based on the transmission SNR condition for the hard data fusion in our scheme. Furthermore, comparing Figure 7 with Figure 4, it can be found that the AFC's performance of the hard fusion outperformed the soft one in the whole SNR interval considered. This was contributed to by the ideal channel information used and the constellation rotation being compensated perfectly. However, the decreasing speed of the error probability at a high SNR was lower than for the soft fusion. Under the influence of noise, some dormant sensing nodes' signals were mistakenly included in final data fusion. Their unreliability confused the data fusion and this problem became more prominent for a high SNR.

In addition, Figure 8 demonstrates clearly that one-bit and two-bit quantization schemes had the same AFC error performance at an SNR of 5 dB in the whole $\beta$ range from 0.4 to 1, while two-bit quantization outperformed the other two cases when the SNR was large, especially for a relative low $\beta$. Moreover, the inflection phenomenon of the AFC error versus $\beta$ became more obvious for the hard data fusion than for the soft data fusion. This told us that preventing the sensing nodes with low-reliability local decisions to send data was more necessary for the hard fusion.

Figure 9 also gives the rotation angle's effect on the performances of the AFC and EFC with the hard fusion. As for the EFC, its error performance still reduced with $\phi$ raising, while the AFC's error almost did not change with $\phi$. This was because the constellation rotation was compensated perfectly by the hard decision at the AFC with the ideal channel amplitude, no matter what the rotation angle was.

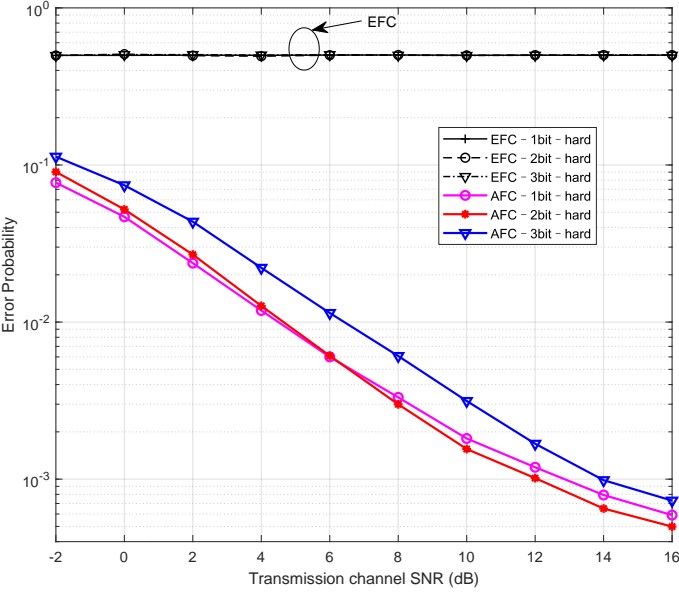

**Figure 7.** Error probabilities of the hard data fusion varying with the SNR of transmission channel for $\beta = 0.8$ and $snr_L = 5$ dB.

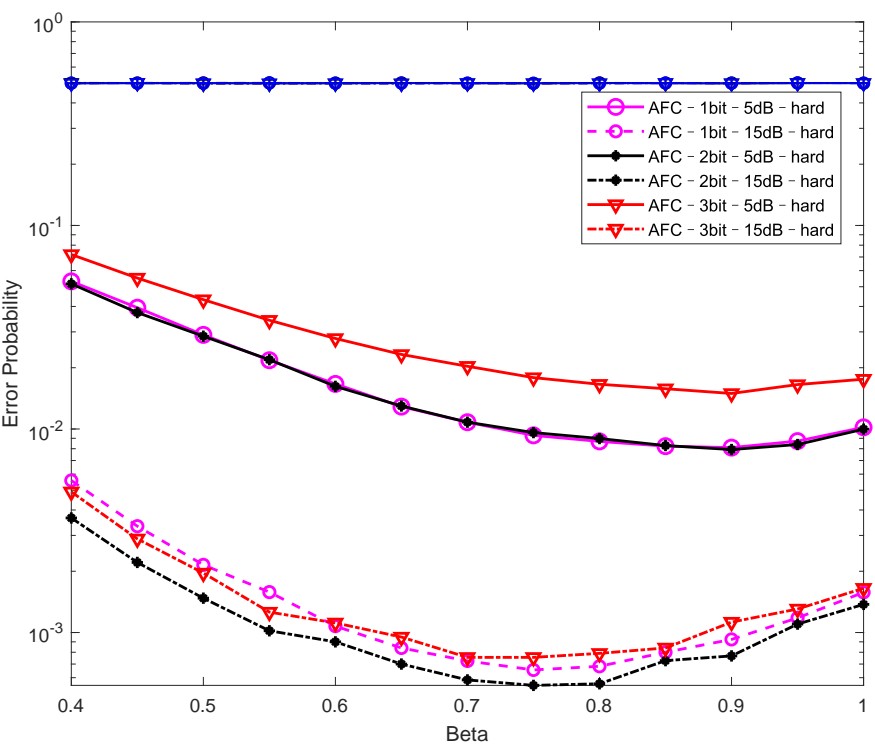

**Figure 8.** Error probabilities of the hard data fusion varying with $\beta$ for SNR = 5, 15 dB and $snr_L$ = 5 dB.

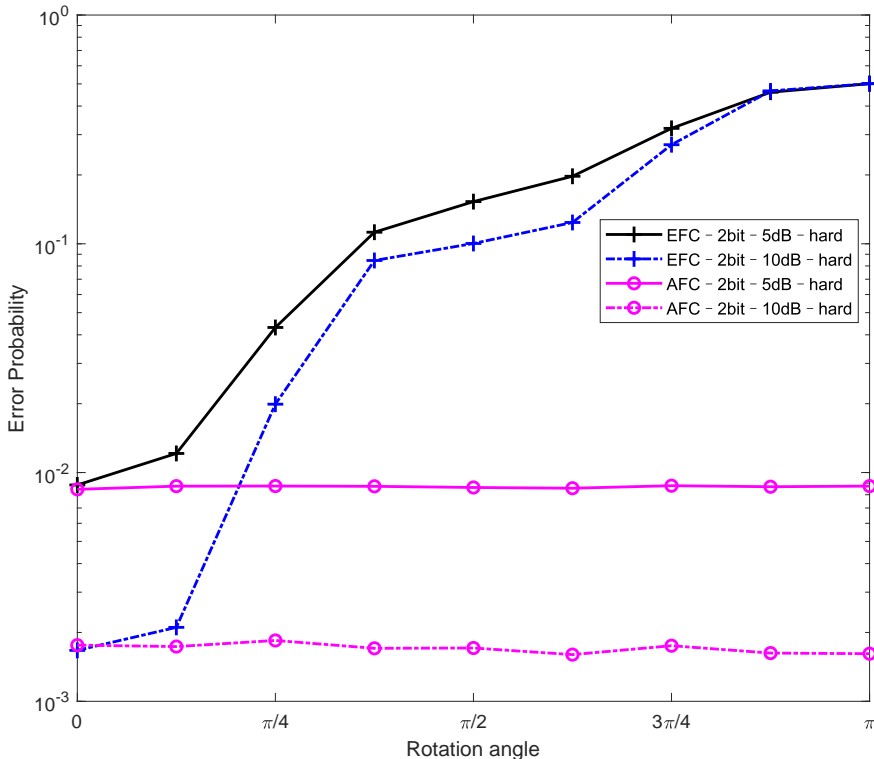

**Figure 9.** Error probabilities of the hard data fusion varying with $\phi$ for SNR = 5, 10 dB and $snr_L$ = 5 dB.

### 5.3. Optimization of Rotation Angle under Error Constraint of EFC

Figure 10 shows the optimal constellation rotation angle $\phi$, which varied with the given constraint of the EFC's error probability under the case of a soft data fusion. For each error constraint within $[0.1, 0.5]$, the optimal $\phi$ was searched. It can be seen that perfect security was realized when the constellation rotation angle was set as $\pi$ for three cases. Moreover, for the same $P_{th}^E$, a larger $\phi$ was needed for a higher SNR. Compared with the three-bit quantization case, a larger $\phi$ was needed under the two-bit quantization case.

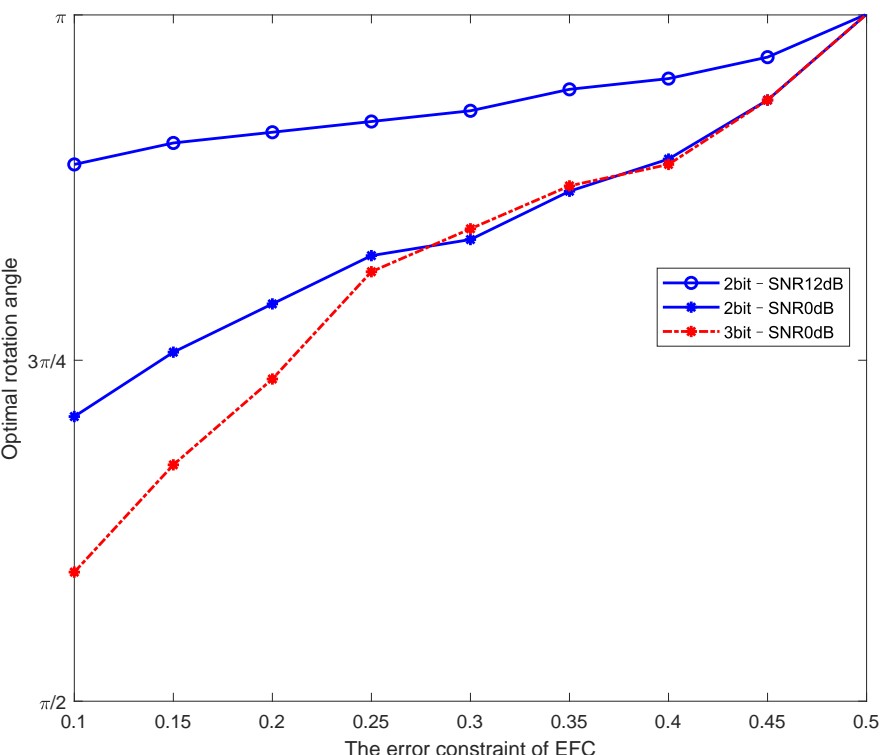

**Figure 10.** The optimal rotation angle vs error constraint of EFC $P_{th}^E$ for $snr_L = 5$ dB.

### 5.4. Comprehensive Comparisons of Various Algorithms

Table 3 compares the proposed scheme with the baseline methods comprehensively and lists some key results about the computational complexity, channel state information (CSI) requirements and the error performance of the two fusion centers. The computational complexity of the kind of schemes with SDF mainly depends on the high computation cost of the numerical integration included in Equations (7) and (8). Moreover, the higher the order of the quantization, the higher the number of numerical integrations. From Table 3 and Figure 4, it can be seen that the proposed scheme with two-bit quantization and SDF obtained a relative better AFC error performance at the cost of a moderate complexity and a moderate CSI requirement, under the perfect security constraint. Although the proposed scheme with HDF could achieve the best AFC error performance with the lowest complexity, it needed the instantaneous information about both the amplitude and phase of the channel. The CSI requirement was the highest. In addition, with the quantization order increasing to three, the computation overhead of the schemes with SDF improved significantly; however, the corresponding AFC error performance reduced conversely, just as shown in Figure 4. Hence, two-bit quantization is preferred for applying the proposed scheme.

**Table 3.** Performance comparison of different schemes in the case of perfect security.

| Scheme | Computational Complexity | CSI Requirement | Error Probabilities of EFC | Error Probabilities of AFC | | | |
|---|---|---|---|---|---|---|---|
| | | | | Low $\beta$ | | High $\beta$ | |
| | | | | Low SNR | High SNR | Low SNR | High SNR |
| Scheme 1 with 1-bit quantization and Sdf [25] | Low | Statistical amplitude | 0.5 | – | – | High | Moderate |
| Expanded version of scheme 1 with 2-bit quantization and Sdf | Moderate | Statistical amplitude and instantaneous phase | 0.5 | High | Moderate | Moderate | Low |
| Expanded version of scheme 1 with 3-bit quantization and Sdf | High | Statistical amplitude and instantaneous phase | 0.5 | – | – | Moderate | Low |
| Scheme 2 with 1-bit quantization and Sdf [26] | Low | Statistical amplitude | 0.5 | High | Moderate | High | Moderate |
| Proposed scheme with 2-bit quantization and Sdf | Moderate | Statistical amplitude and instantaneous phase | 0.5 | High | Moderate | Moderate | Low |
| Proposed scheme with 3-bit quantization and Sdf | High | Statistical amplitude and instantaneous phase | 0.5 | – | – | Moderate | Low |
| Proposed scheme with 1-bit quantization and Hdf | Low | Instantaneous amplitude and phase | 0.5 | Moderate | Low | Moderate | Low |
| Proposed scheme with 2-bit quantization and Hdf | Low | Instantaneous amplitude and phase | 0.5 | Moderate | Low | Moderate | Low |
| Proposed scheme with 3-bit quantization and Hdf | Low | Instantaneous amplitude and phase | 0.5 | Moderate | Low | Moderate | Low |

## 6. Conclusions

In this paper, a random-constellation-rotation-based secure distributed detection under a specific energy constraint was proposed so that the case that a multiple-bit quantization would be covered in local sensing. Moreover, the optimization of local decision thresholds and the rotating threshold were analyzed from the perspective of perfect security. Then, the optimization of the constellation rotation angle was discussed under a flexible constraint on the EFC's error probability. The proposed scheme extended the JLDWT method to a more general case with any multiple-bit local decision. The simulation results demonstrated that a better AFC performance could be obtained by adopting two-bit quantization under the perfect security condition for both soft and hard data fusions. Specifically, at the AFC error probability of $10^{-2}$, the SNR gain of the two-bit local decision against the one-bit case was greater than 3 dB under the conditions of perfect security, $\beta = 8$ and soft data fusion. Compared with the three-bit case, there was also a performance gain of about 0.5

dB. Moreover, through optimizing the constellation rotation angle under a relaxed security constraint, a better error performance for the AFC could be obtained under the soft fusion case. The proposed scheme is recommended to be utilized in the wireless distributed ISAC scenario for 6G networks. It can be helpful for enhancing the secure data fusion under a strict system energy constraint.

Due to the extremely complex expression of the EFC's error probability relative to the rotation angle $\phi$, an exhaustive search was used to find the optimized rotation angle. In future work, the analytic expression of the EFC error probability will be studied to simplify the optimization of various parameters. In addition, the proposed scheme was evaluated by computer simulation in this paper. In order to demonstrate its practicability and verify the performance gain shown in simulation, we plan to design and exploit an experiment system based on software radio platforms combined with hardware sensing nodes in future work.

**Author Contributions:** Conceptualization, G.Z.; methodology, G.Z. and H.S.; software, H.S., J.Y. and G.Z.; writing—original draft preparation, G.Z. and H.S.; writing—review and editing, G.Z. and J.Y. All authors have read and agreed to the published version of the manuscript.

**Funding:** This research was funded by State Key Laboratory of Geo-information Engineering of China under Grant Number (no. SKLGIE2020-Z-2-l).

**Data Availability Statement:** Data will be available based on reasonable request to the corresponding authors.

**Conflicts of Interest:** The authors declare no conflict of interest.

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
