# Peer review of "Secure Decision Fusion in ISAC-Oriented Distributed Wireless Sensing Networks with Local Multilevel Quantization"

_electronics, doi:10.3390/electronics12061428_

Round 1

Reviewer 1 Report

In this manuscript, the authors describe a secure distributed decision fusion framework that involves making local (like federated) decisions based on multiple bits quantization and transmitting the signals using constellation rotation. This framework has also the requirement of being energy-efficient for the whole integrated sensing and communication system. The authors present their methodology focusing on the energy efficiency aspect, as well as on the anti-eavesdropping scheme that involves the rotation of the modulation constellation. Furthermore, they evaluate the proposed framework via simulation results that eventually validate their method. The presented work is interesting and the references are appropriate, permitting a smooth follow-up of the problem. No comments on the methodology and the presented results – just carefully proofread your manuscript for typos before publication (e.g. page 4, line 121 phrase instead of phase).

Reviewer 2 Report

This article presents a very interesting approach for distributed wireless sensing. The authors should consider the following remarks:

1) The abstract section is too long. Please shorten it.

2) Please specify which simulation tool is used at the simulation results section.

3) At the end of section 1 please explain clearly the novelty of the paper compared to published literature.

4) Please include the main numerical research findings at abstract and conclusions sections.

5) At conclusions section please specify where the proposed approach can be implemented (6G systems?).

6) At Figure 3 clearly specify what x and y axes represent.

7) A few acronyms, like JLDWT and SNR are not explained.

8) The analysis of the simulation results is adequate, but please be clear why the proposed scheme is superior to already existing ones.

Reviewer 3 Report

The work examines physical layer security for distributed sensing toward ISAC. In particular, the authors propose a secure distributed decision fusion scheme combining multi-bit quantization-based local decisions and constellation rotation-based transmission. The work is interesting and poses significance in the domain. However, revisions are required before the work can be processed further.

1. Highlight the key contributions of this paper in bullet form and add the paper organization in section 1.

2. The motivation for this work needs to be critically justified.

3. The authors need to clarify why the AFC’s fusion performance will decrease conversely due to the worse transmission of a high-order modulation in the main channel for the case of higher-order quantization. The reasons given and the results shown are not convincing enough. A more elegant description of this scenario is requested.

4. The experimental setting is missing in this work. Can you add a test scenario to verify your simulated claims?

5. The analytical description of the results is very good. However, I want to see a complex table showing the key results and comparison with the existing state-of-the-art.

6. I suggest that you add a few more recent references to strengthen the theoretical background of the study. The following paper could be useful to enhance the remark on 6G wireless networks in section 1: https://www.mdpi.com/1424-8220/21/5/1709

Round 2

Reviewer 3 Report

The authors have addressed my earlier concerns sufficiently.